# CONTINUAL PROTOTYPE EVOLUTION: LEARNING ONLINE FROM NON-STATIONARY DATA STREAMS

## ABSTRACT

Attaining prototypical features to represent class distributions is well established in representation learning. However, learning prototypes online from streams of data proves a challenging endeavor as they rapidly become outdated, caused by an ever-changing parameter space in the learning process. Additionally, continual learning does not assume the data stream to be stationary, typically resulting in catastrophic forgetting of previous knowledge. As a first, we introduce a system addressing both problems, where prototypes evolve continually in a shared latent space, enabling learning and prediction at any point in time. In contrast to the major body of work in continual learning, data streams are processed in an online fashion, without additional task-information, and an efficient memory scheme provides robustness to imbalanced data streams. Besides nearest neighbor based prediction, learning is facilitated by a novel objective function, encouraging cluster density about the class prototype and increased inter-class variance. Furthermore, the latent space quality is elevated by pseudo-prototypes in each batch, constituted by replay of exemplars from memory. We generalize the existing paradigms in continual learning to incorporate *data incremental learning* from data streams by formalizing a two-agent *learner-evaluator* framework, and obtain state-of-the-art performance by a significant margin on eight benchmarks, including three highly imbalanced data streams.

## 1 INTRODUCTION

The prevalence of data streams in contemporary applications urges systems to learn in a continual fashion. Autonomous vehicles, sensory robot data, and video streaming yield never-ending streams of data, with abrupt changes in the observed environment behind every vehicle turn, robot entering a new room, or camera cut to a subsequent scene. Alas, learning from streaming data is far from trivial due to these changes, as neural networks tend to forget the knowledge they previously acquired. The data stream presented to the network is not identically and independently distributed (iid), emanating a trade-off between neural stability to retain the current state of knowledge and neural plasticity to swiftly adopt the new knowledge (Grossberg, 1982). Finding the balance in this stability-plasticity dilemma addresses the catastrophic forgetting (French, 1999) induced by the non-iid intrinsics of the data stream, and is considered the main hurdle for continually learning systems.

Although a lot of progress has been established in the literature, often strong assumptions apply, impeding applicability for real-world systems. The static training and testing paradigms prevail, whereas a true continual learner should enable both simultaneously and independently. Therefore, we propose the two-agent *learner-evaluator* framework to redefine perspective on existing paradigms in the field. Within this framework, we introduce *data incremental learning*, enabling completely task-free learning and evaluation.

Furthermore, we introduce Continual Prototype Evolution (CoPE), a new online data incremental learner wherein prototypes perpetually represent the most salient features of the class population, shifting the catastrophic forgetting problem from the full network parameter space to the lower-dimensional latent space. As a first, our prototypes evolve continually with the data stream, enabling learning and evaluation at any point in time. Similar to representativeness heuristics in human cognition (Kahneman & Tversky, 1972), the class prototypes are the cornerstone for nearest neighbor classification. Additionally, the system is robust to highly imbalanced data streams by the combination

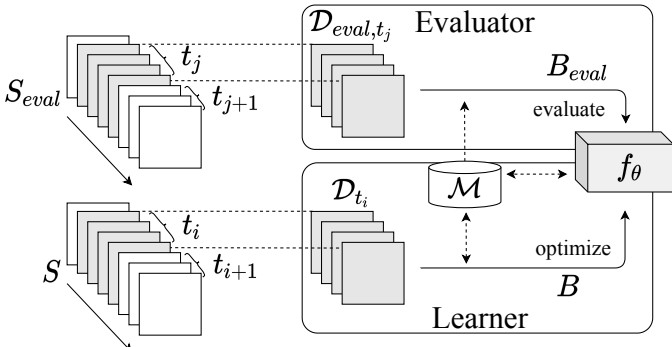

Figure 1: Overview of the *learner-evaluator* framework, overcoming the static training and testing paradigms by explicitly modelling continual optimization and evaluation from data streams in the *learner* and *evaluator* agents. The framework generalizes to both continual learning and concept drift with resources transparently defined as the horizon $\mathcal{D}$ and operational memory $\mathcal{M}$.

of replay with a balancing memory population scheme. We find batch information in the latent space to have a significant advantage in the challenging non-stationary and online processing regime, which we incorporate in the novel pseudo-prototypical proxy loss.

## 2 THE LEARNER-EVALUATOR FRAMEWORK

To date, the paradigms of task, class, and domain incremental learning (van de Ven & Tolias, 2018) dominate the continual learning literature. However, strong and differing assumptions often lead to confusion and overlap between implementations of these definitions. Furthermore, the concept of a static training and testing phase is still ubiquitous, whereas continual learning systems should enable both phases continually and independently. Therefore, we propose a generalizing framework which disentangles the continually learning system into two agents: the *learner* and the *evaluator*. Figure 1 presents an overview of the framework.

The learning agent learns predicting function $f_\theta : \mathcal{X} \rightarrow \mathcal{Y}$ parameterized by $\theta$, mapping the input space $\mathcal{X}$ to the target output space $\mathcal{Y}$. The learner receives data samples $(\mathbf{x}_i, \mathbf{y}_i)$ from stream $S$ and has simultaneous access to the *horizon* $\mathcal{D}$, i.e. the observable subset of stream $S$ which can be processed for multiple iterations. Data sample $i$ is constituted by input feature $\mathbf{x}_i \in \mathcal{X}$ and corresponding (self-)supervision signal $\mathbf{y}_i$ for which the output space for classification is defined as a discrete set of observed classes $\mathcal{Y}_i \leftarrow \mathcal{Y}_{i-1} \cup \{\mathbf{y}_i\}$. To manage memory usage and to enable multiple updates and stochasticity in the optimization process, updates for $\theta$ are typically performed based on a small-scale *processing batch* $B \subseteq \mathcal{D}$. The data and size of the horizon $\mathcal{D}$ are determined by the specific setup or application, ranging from standard offline learning with $\mathcal{D} = S$ to online continual learning with $\mathcal{D} = B$. Furthermore, the learner might need additional resources after observing data from $B \subseteq \mathcal{D}$, such as stored samples or model copies, confined by the *operational memory $\mathcal{M}$*.

The evaluating agent acts independently from the learner by evaluating $f_\theta$ with horizon $\mathcal{D}_{eval}$ from the evaluation stream $S_{eval}$, with small-scale processing batches $B_{eval} \subseteq \mathcal{D}_{eval}$. This stream can contain yet unobserved concepts by the learner in $S$ to measure zero-shot performance. The framework provides leeway for the concept distributions in $S_{eval}$ being either static or dynamically evolving, determining how performance of the learner is measured. On the one hand, static concept distributions can measure the degree to which the knowledge of learned concepts is preserved, as commonly used in continual learning. On the other hand, evolving concept distributions measure performance for the current distribution in horizon $\mathcal{D}_{eval}$ only, where concepts might drift from their original representation, also known as concept drift (Schlimmer & Granger, 1986). Evaluation can occur asynchronously on-demand or periodically with periodicity $\rho$ determining the resolution of the evaluation samples.

**Task, class, and domain incremental learning** are based on the composition in the learner for the observable stream subset in horizon $\mathcal{D}_t$, which is incrementally replaced by a new subset of data

for the new task, set of classes, or domain, with $t$ the identifier of the present data subset. Task incremental learning assumes both learner and evaluator to get data $(\mathbf{x}_i, \mathbf{y}_i, t_i)$ with $t_{i+1} \geq t_i$ and the horizon spanning all data of a given task with $\mathcal{D}_t = \{(\mathbf{x}_i, \mathbf{y}_i, t_i) \in S \mid t_i = t\}$ (De Lange et al., 2019; van de Ven & Tolias, 2019). Having explicit access to $t_i$ confines prediction to an isolated output space. Similarly, in class incremental learning the learner implicitly requires $t_i$ to identify the transitions of $\mathcal{D}$, when observing new batches of classes (Rebuffi et al., 2017; Castro et al., 2018; Shmelkov et al., 2017; Wu et al., 2018). However, the evaluator considers the entire output space without the need for identifier $t$. Domain incremental learning holds the same assumptions as class incremental learning, with concepts drifting from one domain to the other for a typically fixed output space, exemplified by the widely used permuted-MNIST setup (Goodfellow et al., 2013).

**Data incremental learning** is a more general paradigm we introduce to facilitate learning from any data stream, with no assumption but to observe data incrementally. In contrast to existing paradigms, when the learner observes horizon $\mathcal{D}$ of data stream $S$, data incremental learning does not disclose an identifier $t$. Consequently, there is no explicit indication to which subset of the stream is being observed in the horizon $\mathcal{D}$. Therefore, the learner either processes observed data directly in an online fashion with processing batch $B = \mathcal{D}$, or infers an implicit identifier $t$ from statistics in stream $S$. Similar to class and domain incremental learning, the evaluator operates without $t$ on the full output space. This paradigm endows continually learning systems with increased practical use, as real-world streaming applications often lack supervision signal $t$. Moreover, even if $t$ is provided, this would introduce a bias in the fixed choice of the supervisor, rather than dynamically determined based on the needs of the system.

## 3 PRIOR WORK

Continually learning systems are able to learn with limited resources from data streams prone to severe distribution shifts. The main body of works presumes the presence of tasks, which divide the data streams into large discrete batches, and are indicated to the learner with a task identifier (Kirkpatrick et al., 2017; Li & Hoiem, 2017; Zenke et al., 2017; Aljundi et al., 2018; De Lange et al., 2020). Replay methods retain representative data for observed data distributions, currently unavailable in the learner's horizon $\mathcal{D}$. The replay data is either obtained directly from operational memory $\mathcal{M}$ with stored samples (Rebuffi et al., 2017; Lopez-Paz & Ranzato, 2017) or generated using generative models (Shin et al., 2017; Kamra et al., 2017; Seff et al., 2017; Wu et al., 2018). **GEM** (Lopez-Paz & Ranzato, 2017) uses replay in a constraint optimization perspective to project gradients towards a local joint task optimum. **iCaRL** (Rebuffi et al., 2017) employs exemplars to distill knowledge (Hinton et al., 2015) to the learner from a previous model version, with new class exemplars stored in a queue to optimally represent the class mean in feature space. The prototypes are then used for nearest neighbor prediction by the evaluator, in the same vein as concurrent work to ours (Han et al., 2020). Nonetheless, all three works strongly rely on task identifier $t$ for the learner, mostly unavailable for real-world data streams. Moreover, in both prototypical approaches the prototypes remain static between the given task transitions and become outdated. Consequently, before using the evaluator they have to exhaustively recalculate the prototypes based on all exemplars in memory. In contrast, our prototypes evolve in an online fashion with the data stream and remain representative for the continual learner and evaluator at all times.

Recent works focus on online data incremental learning (Section 2) in which the learner operates completely task-free. **Reservoir** (Vitter, 1985) is a replay baseline with strong potential to outperform continual learning methods (Chaudhry et al., 2019). Samples are stored in memory $\mathcal{M}$ with probability $M/n$, with $n$ the number of observed samples and buffer size $M$. **MIR** (Aljundi et al., 2019a) extends Reservoir sampling with a loss-based retrieval strategy, with the cost of additional forward passes and a model copy to attain the losses for a subset of samples. The Reservoir buffer population approximately follows the data stream distribution, severely deteriorating the performance of underrepresented tasks in imbalanced data streams, as shown in Section 6.2. An alternative memory population scheme is used in **GSS** (Aljundi et al., 2019b) by extending the GEM constraint optimization perspective to an instance-based level. GSS adds samples to the buffer based on their gradients, whereas GEM requires the number of tasks and the task transitions to divide memory equally over all tasks a priori. In contrast, iCaRL's memory population is incrementally subdivided over all classes after learning a task, by iteratively adding observed samples from $\mathcal{D}$ to optimally approximate the class mean in feature space. As this is computationally expensive, concurrent works to ours explore other balancing

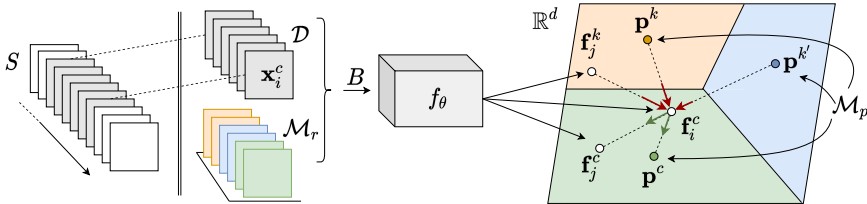

Figure 2: Main setup. The learner updates network $f_\theta$ and prototypes $\mathbf{p}^y, \forall y \in \mathcal{Y}$ continually. The PPP-loss encourages inter-class variance (red arrows) and reduces intra-class variance (green arrows).

schemes (Kim et al., 2020; Chrysakis & Moens, 2020), where we propose a simple but effective class-based Reservoir scheme with uniform retrieval.

Another branch of parameter isolation methods (De Lange et al., 2019) allocates parameters to subsets of the data. Several task incremental works assign parameters based on the task identifier (Mallya & Lazebnik, 2018; Serra et al., 2018). A new line of work instead focuses on task-free model expansion. **CURL** (Rao et al., 2019) enables task-free and unsupervised adaptation using a multi-component variational auto-encoder, with generative replay from a model copy avoiding forgetting in the current model. **CN-DPM** (Lee et al., 2020) allocates data subsets to expert networks following a Dirichlet process mixture. In contrast to these capacity expansion based methods, CoPE evades unbound allocation of resources, as the memory and network capacity are fixed with the replay memory dynamically subdivided over categories occurring in the data stream. Note that new categories require an additional prototype, but these are only $d$-dimensional and therefore insignificant in size, and the set of categories is typically limited as well.

Besides the focus on continual learning in this work, our learner-evaluator framework generalizes to concept drift as well (Schlimmer & Granger, 1986), for which we refer to an overview in (Tsymbal, 2004; Gama et al., 2014). Further, in deep embedding learning most commonly pairs (Hadsell et al., 2006) and triplets (Harwood et al., 2017) of samples are considered in contrastive losses, whereas other works use batch information in lifted structure embeddings (Oh Song et al., 2016), or instance-wise softmax embeddings (Ye et al., 2019). These approaches fully depend on the batch size, whereas our pseudo-prototypical proxy loss aggregates both decoupled prototypes and the additional batch pseudo-prototypes to defy class interference in the latent space. Learning prototypical representations also shows promising results in few-shot learning (Snell et al., 2017).

## 4 CONTINUAL PROTOTYPE EVOLUTION

The online data incremental learning setup of the learner is described in Figure 2. Embedding network $f_\theta$ maps processing batch $B$, composed of samples in horizon $\mathcal{D}$ from the non-iid data stream $S$ and operational memory $\mathcal{M}$, to low-dimensional $\mathbb{R}^d$ latent space, followed by a nearest neighbor classifier. We enforce $||f_\theta(\mathbf{x}_i)|| = 1$ with an L2 normalization layer. $\mathcal{M}$ is subdivided in a replay memory $\mathcal{M}_r$ and prototypical memory $\mathcal{M}_p$. CoPE comprises three main components: continually evolving representations, balanced replay and the pseudo-prototypical proxy (PPP) loss. In the following, we discuss these components and formalize the optimal choice of prototype, with $\mathbf{f}_i^c$ denoting latent space projection $f_\theta(\mathbf{x}_i^c)$ for an instance $\mathbf{x}_i$ of class $c$. For the full algorithm, we refer to Appendix A.

### 4.1 EVOLVING REPRESENTATIONS

Each observed class $c \in \mathcal{Y}$ is represented by a slowly progressing prototype $\mathbf{p}^c$ in operational memory $\mathcal{M}_p$. The nearest neighbor classifier finds the most similar prototype for the given query $\mathbf{x}_i$, predicting $c^* = \arg\max_{c \in \mathcal{Y}} \mathbf{f_i^T p^c}$. Similar to (Mensink et al., 2013; Rebuffi et al., 2017), the class-prototype approximates the center of mass in the latent space, which we formally justify in Section 4.4. The main crux with storing representations is to prevent them from becoming obsolete as the embedding network evolves. Additionally, this is further complicated by the shifting data distributions in the non-stationary regime, incurring catastrophic forgetting. Experience replay from a buffer $\mathcal{M}_r$ is a well known approach to address this forgetting. Nonetheless, in our setup the replayed exemplars gain additional information about the current state of the embedding space, enabling rehearsal to rectify approximation $\mathbf{p}^c$ to the true center of mass. Concretely, the sampled batch

$B_n$ equals the horizon $\mathcal{D}$ from data stream $S$ and joins batch $B_{\mathcal{M}}$ of equal size from memory $\mathcal{M}_r$, constituting $B$ as $B_n \cup B_{\mathcal{M}}$. However, updating the prototypes by fully relying on features extracted from $B$ incurs an unstable optimization process as the representative prototypes depend on stochastic sampling of the class distributions. Therefore, we design the prototypes to evolve continually with a high momentum based update for each observed batch, aiming to stabilize the impetuous changes in the data stream:

$$\mathbf{p}^c \leftarrow \alpha\mathbf{p}^c + (1-\alpha)\bar{\mathbf{p}}^c, \text{ s.t. } \quad \bar{\mathbf{p}}^c = \frac{1}{|B^c|}\sum_{\mathbf{x}^c \in B^c} f_\theta(\mathbf{x}^c), \tag{1}$$

with momentum parameter $\alpha \in [0,1]$, the batch subset $B^c = \{(\mathbf{x}_i, y_i = c) \in B\}$ of class $c$, and $\bar{\mathbf{p}}^c$ the corresponding center of mass in latent space for the current batch. Due to triangle inequality $\mathbf{p}^c$ is no longer unit length and requires to be L2-normalized after the update in Eq. 1. We empirically validate the effectiveness of high momentum with $\alpha \approx 1$ in the ablation study in Appendix D.

## 4.2 BALANCED REPLAY

Similar to Rebuffi et al. (2017); Chrysakis & Moens (2020), the total buffer size $M$ is equally divided over the number of observed classes $|\mathcal{Y}|$ in a dynamic fashion. This scheme ensures consistent buffer capacity for all classes, making memory allocation independent of the data stream characteristics. As $S$ is typically highly imbalanced in real-world scenarios, this memory scheme prevents classes to be eradicated from the buffer and assumes equal importance to represent each class at all times. Consequently, random retrieval from the buffer resembles sampling an iid replay batch. Furthermore, each class-specific replay memory $\mathcal{M}_r^c$ can simply capture a random subset of its parent class distribution to approximate its center of mass. This avoids computationally expensive herding techniques as in iCaRL (Rebuffi et al., 2017), which would require recalculation of the feature means on each change of the memory size or network parameters.

## 4.3 PSEUDO-PROTOTYPICAL PROXY LOSS

The learner optimizes $f_\theta$ to project an instance $\mathbf{f}_i^c \in \mathbb{R}^d$ of class $c$ close to its corresponding prototype $\mathbf{p}^c$ in the latent space. As the prototype acts as a surrogate for the class mean in latent space, the cluster population has a common reference point to reduce intra-class variance, and enforce inter-class variance by remaining distant from the other class prototypes. Additionally, due to the embedding architecture we can use intrinsic information of the batch samples in the latent space. Therefore, we exploit the supervision signal $\mathbf{y}_i$ in a sample $(\mathbf{x}_i, \mathbf{y}_i) \in B$ not only to indicate which class $\mathbf{x}_i$ belongs to, but also to make the distinction between positive and negative pairs in $B$. Consequently, we can define one-against-all subsets for an instance of class $c$, with positives from the same class in $B^c = \{(\mathbf{x}_i, y_i = c) \in B\}$ and negatives in $B^k$. Starting from these sets, the prototypical attractor and repellor sets for an instance $\mathbf{x}_i^c$ are constituted with the class prototype $\mathbf{p}^c$ and the other instances in $B$. First, the other instances of class $c$ act as pseudo-prototypes $\hat{\mathbf{p}}^c$ in attractor set $\mathbb{P}_i^c = \{\mathbf{p}^c\} \cup \{\hat{\mathbf{p}}_j^c = f_\theta(\mathbf{x}_j^c) \mid \forall \mathbf{x}_j^c \in B^c, i \neq j\}$. Second, the samples of other classes $\mathbf{x}_j^k \in B^k$ should instead avoid both $\mathbf{x}_i^c$ in latent space and the class representative $\mathbf{p}^c$, defined by repellor set $\mathbb{U}_i^c = \{\mathbf{p}^c, \hat{\mathbf{p}}_i^c = f_\theta(\mathbf{x}_i^c)\}$. The attractor set for $\mathbf{x}_i^c$ facilitates a decrease in intra-class variance with $\mathbf{p}^c$ safeguarding the absence of positive batch pairs with $1 \leq |\mathbb{P}_i^c| \leq |B^c|$, whereas the repellor exploits $\mathbf{x}_i^c$ and corresponding prototype as a reference point to increase inter-class variance. To incorporate the attractor and repellor sets, we formulate a binary classification problem similar to Ye et al. (2019), with the joint probability that instance $\mathbf{x}_i^c$ is predicted as class $c$ and instances $\mathbf{x}_j^k \in B^k$ not being predicted as class $c$

$$P_i = P(c|\mathbf{x}_i^c)\prod_{\mathbf{x}_j^k}(1 - P_i(c|\mathbf{x}_j^k)) \tag{2}$$

with the assumption of independence between $\mathbf{x}_i^c$ and $\mathbf{x}_j^k$ being recognized as $c$. We define the expected posterior probabilities for the attractor and repellor sets of instance $\mathbf{x}_i^c$ respectively as

$$P(c|\mathbf{x}_i^c) = \mathbb{E}_{\tilde{\mathbf{p}}^c \in \mathbb{P}_i^c}[P(c|\mathbf{f}_i^c, \tilde{\mathbf{p}}^c)], \quad P_i(c|\mathbf{x}_j^k) = \mathbb{E}_{\tilde{\mathbf{p}}^c \in \mathbb{U}_i^c}\left[P(c|\mathbf{f}_j^k, \tilde{\mathbf{p}}^c)\right], \tag{3}$$

with $\tilde{\mathbf{p}}^c$ a proxy for the latent mean of class $c$ in

$$P(c|\mathbf{f}, \tilde{\mathbf{p}}^c) = \frac{\exp(\mathbf{f}^T\tilde{\mathbf{p}}^c/\tau)}{\exp(\mathbf{f}^T\tilde{\mathbf{p}}^c/\tau) + \sum_{k \neq c}\exp(\mathbf{f}^T\mathbf{p}^k/\tau)}, \tag{4}$$

where temperature $\tau$ controls the concentration level of the distribution (Hinton et al., 2015), assuming a cosine similarity metric $\mathbf{f}_i^T \mathbf{f}_j$ with vectors normalized to unit length. We reformulate the objective in Eq.(2) as loss function $\mathcal{L}$ by negative log-likelihood and summation over all the instances in $B$, which approximates the true joint probability with assumed independent pairs in the batch:

$$\mathcal{L} = -\frac{1}{|B|} \left[ \sum_i \log P(c|\mathbf{x}_i^c) + \sum_i \sum_{\mathbf{x}_j^k} \log(1 - P_i(c|\mathbf{x}_j^k)) \right]. \tag{5}$$

### 4.4 OPTIMAL PROTOTYPES

We update prototypes to approximate the mean of the parent distribution in Eq.(1). This assumption is optimal for Bregman divergences for which the cluster mean is shown to have minimal distance to its population (Banerjee et al., 2005). This Bregman divergence is defined for a differentiable, strictly convex function $\varphi$ as

$$d_\varphi(\mathbf{f}_i, \mathbf{f}_j) = \varphi(\mathbf{f}_i) - \varphi(\mathbf{f}_j) - (\mathbf{f}_i - \mathbf{f}_j)^T \nabla \varphi(\mathbf{f}_j), \tag{6}$$

for which the squared Euclidean distance with $\varphi(\mathbf{f}) = ||\mathbf{f}||^2$ is a canonical example. The squared Euclidean distance is proportional to the cosine distance with vectors normalized to unit length: $\frac{1}{2}||\mathbf{f}_i - \mathbf{f}_j||^2 = 1 - \cos \angle(\mathbf{f}_i, \mathbf{f}_j)$. As the PPP-loss in Eq.(4) requires a similarity measure instead of a distance measure, we employ the complementary normalized cosine similarity $\cos \angle(\mathbf{f}_i, \mathbf{f}_j) = \mathbf{f}_i^T \mathbf{f}_j$ with $||\mathbf{f}_i|| = ||\mathbf{f}_j|| = 1$. Besides the desirable cluster-mean property of its complement, this metric is also efficient for calculating the full batch similarity matrix using matrix multiplication libraries.

## 5 EXPERIMENTS

This work examines five balanced data streams and 15 highly imbalanced variants based on Split-MNIST, Split-CIFAR10 and Split-CIFAR100, from which two low-capacity balanced setups are discussed in Appendix E. The learner is presented a data stream $S$, constituted by a sequence of tasks, each delineated by a subset of classes from the original dataset. Although the learner in CoPE is completely ignorant to the notion of task, this setup enables comparing to methods requiring task boundaries such as GEM and iCaRL. The evaluator uses a held-out dataset of static concepts in $S_{eval}$, evaluating with the subset of seen concepts $\mathcal{Y}$ in $\mathcal{D}_{eval}$ using the accuracy metric. The CoPE learner processes data online with $B_n = \mathcal{D}$ in the data incremental setup, allowing per-task processing of 1 epoch for methods requiring task boundaries with $B \subset \mathcal{D}$. We use vanilla stochastic gradient descent with a limited processing batch size $|B_n|$ of 10 as in in (Lopez-Paz & Ranzato, 2017; Aljundi et al., 2019b; Lee et al., 2020). All results are averaged over 5 different network initializations.[1]

**Balanced data streams** contain a similar amount of data per task. We consider three benchmarks. First, **Split-MNIST** constitutes the MNIST (LeCun et al., 1998) handwritten digit recognition dataset with 60k training samples, split into 5 tasks according to pairs of incrementing digits. Second, **Split-CIFAR10** considers the CIFAR10 (Krizhevsky et al., 2009) dataset, subdivided into 5 tasks with 2 labels each, where each task entails 10k training samples. Third, **Split-CIFAR100** is a variant of the CIFAR dataset with 100 different classes. The 50k training samples are subdivided in 20 tasks of 2.5k samples as in (Lopez-Paz & Ranzato, 2017; Lee et al., 2020). For all datasets the evaluator considers the entire original test subset for $S_{eval}$.

**Imbalanced data streams** introduce a more realistic scenario without equality assumptions on the task durations in $S$ and address the literature mostly balancing the data streams artificially. Besides the imbalanced Split-MNIST setup (Aljundi et al., 2019b), we introduce two novel and more challenging benchmarks based on Split-CIFAR10 and Split-CIFAR100, where data stream $S$ comprises significantly more data in task $T_i$, denoted by $S(T_i)$. Split-MNIST and Split-CIFAR10 have respectively 2k and 4k samples in $T_i$, whereas tasks $T_j$ for $j \neq i$ contain factor 10 less data for five variants $S(T_i)$, $\forall i \in \{1, ..., 5\}$. Split-CIFAR100 defines $T_i$ with 2.5k samples and 1k for the remaining tasks, with variants $i \in \{1, 5, ..., 20\}$.

**Architectures.** MNIST setups use an MLP with 2 hidden layers of 400 units with 2k memories for the balanced setup as in (Hsu et al., 2018; Lee et al., 2020; van de Ven & Tolias, 2019), and 100 units

---

[1]Appendix details the full setup with additional experiments. Code publicly released upon paper acceptance.

with $|\mathcal{M}| = 0.3k$ for the imbalanced setup as in (Aljundi et al., 2019b). CIFAR setups use a slim version of Resnet18 (He et al., 2016) with a 1k memory size for CIFAR10 (Aljundi et al., 2019b; Lee et al., 2020), and 5k for CIFAR100 (Lopez-Paz & Ranzato, 2017).

**Methods** compared to CoPE entail 11 baselines, with details on prior work discussed in Section 3. The upper reference point for performance when relaxing the challenging non-iid feature in continual learning is set by **iid-online & iid-offline**. The learner shuffles the full data stream $S$ to ensure the iid property, for which *iid-online* trains a single epoch and *iid-offline* multiple epochs. In contrast, the **finetune** learner considers non-iid data stream $S$ sequentially, but optimizes solely for the new batch which typically results in worst-case catastrophic forgetting. **CoPE-CE** is a reference point for the merits of a prototypical approach by solely using the CoPE memory and sampling scheme, but with a typical cross-entropy loss and softmax classifier. **GEM** and **iCaRL** are standard replay methods considered in a class incremental setup, with the learner requiring task boundaries. For online data incremental learning, we consider the **reservoir**, **MIR** and greedy **GSS** replay baselines, with **CURL** and **CN-DPM** instead relying on model expansion.

Table 1: The three balanced data stream accuracies (%) with standard deviation over 5 initializations. Expansion-based methods CURL and DN-CPM report results from their original work.

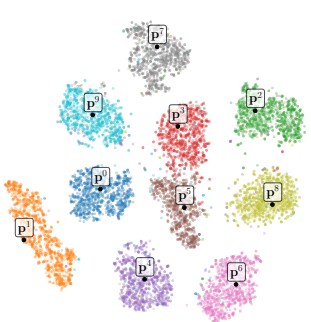

|  | **Split-MNIST** | **Split-CIFAR10** | **Split-CIFAR100** |
|---|---|---|---|
| iid-offline | $98.44 \pm 0.02$ | $83.02 \pm 0.60$ | $50.28 \pm 0.66$ |
| iid-online | $96.57 \pm 0.14$ | $62.31 \pm 1.67$ | $20.10 \pm 0.90$ |
| finetune | $19.75 \pm 0.05$ | $18.55 \pm 0.34$ | $3.53 \pm 0.04$ |
| GEM | $93.25 \pm 0.36$ | $24.13 \pm 2.46$ | $11.12 \pm 2.48$ |
| iCARL | $83.95 \pm 0.21$ | $37.32 \pm 2.66$ | $10.80 \pm 0.37$ |
| CURL (Rao et al., 2019) | $92.59 \pm 0.66$ | − | − |
| DN-CPM (Lee et al., 2020) | $93.23 \pm 0.09$ | $45.21 \pm 0.18$ | $20.10 \pm 0.12$ |
| reservoir | $92.16 \pm 0.75$ | $42.48 \pm 3.04$ | $19.57 \pm 1.79$ |
| MIR | $93.20 \pm 0.36$ | $42.80 \pm 2.22$ | $20.00 \pm 0.57$ |
| GSS | $92.47 \pm 0.92$ | $38.45 \pm 1.41$ | $13.10 \pm 0.94$ |
| CoPE-CE | $91.77 \pm 0.87$ | $39.73 \pm 2.26$ | $18.33 \pm 1.52$ |
| CoPE (ours) | $\mathbf{93.94 \pm 0.20}$ | $\mathbf{48.92 \pm 1.32}$ | $\mathbf{21.62 \pm 0.69}$ |

Figure 3: Balanced Split-MNIST first seed $S_{eval}$ t-SNE (Maaten & Hinton, 2008).

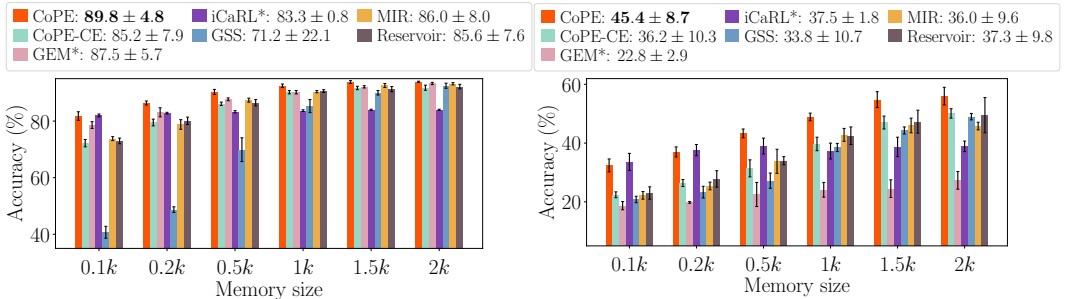

Figure 4: Accuracies over buffer sizes $|\mathcal{M}|$ for balanced Split-MNIST and Split-CIFAR10 sequences. The legend reports averages over all observed buffer sizes. '∗' indicates learner with task information.

## 6 RESULTS AND DISCUSSION

### 6.1 BALANCED DATA STREAMS

The results for the three balanced data streams in Table 1 consistently report state-of-the-art for CoPE. The difficulty for learning online is reflected in the discrepancy of performance between iid-offline and iid-online, indicating increasing difficulty for a minimal 2% for Split-MNIST, raising by factor 10 for Split-CIFAR10, and culminating to 30% in Split-CIFAR100. For Split-MNIST the gap with iid-online performance is closed by 0.7% compared to main competitors GEM and DN-CPM, with

our representations visualized in Figure 3. Furthermore, in the more challenging Split-CIFAR10 setup we significantly increase the gained margin by 3.7%. In the most challenging Split-CIFAR100, CN-DPM, Reservoir and MIR are able to perform on par with the iid-online baseline, however, CoPE establishes an improvement of at least 1.5% over all four baselines.

Compared to balanced replay with standard cross-entropy (CoPE-CE), the prototypical approach (CoPE) proves effective with significant gains of 2.2%, 9.2% and 3.3% respectively over the three benchmarks. Except for GEM in Split-MNIST, class incremental learning methods GEM and iCaRL are not competing in the online setting and additionally require from the setup to reveal an identifier $t$ to the learner. From the expansion-based methods DN-CPM is competitive, whereas CURL is more suited for unsupervised learning and lacks behind. Although Reservoir and extension MIR perform on par with iid-online for Split-CIFAR100, the imbalanced experiments in Section 6.2 show that full reservoir-based population of the buffer strongly relies on this assumption of equally sized tasks, which is unlikely to occur in real-world data streams.

**Buffer size ablation study** in Figure 4 shows CoPE to prevail over all sizes of replay buffer $\mathcal{M}_r$ compared to other replay methods, extending robustness to low capacity regimes. Although iCaRL shows competitive results for low capacity, CoPE scales with growing capacity leading to significantly outperforming iCaRL with 11% in Split-MNIST ($2k$) and Split-CIFAR100 ($5k$), and 17% in Split-CIFAR10 ($2k$). We refer to Appendix E for the Split-CIFAR100 results.

## 6.2 IMBALANCED DATA STREAMS

Results for the highly imbalanced data stream benchmarks are reported in Figure 4. CoPE significantly outperforms all baselines in the three scenarios, with low standard deviation for the 15 variants indicating robustness over a wide spectrum of imbalanced sequences. Gradient-based sample selection (GSS) outperforms Reservoir and MIR for Split-MNIST, in correspondence with results in Aljundi et al. (2019b), whereas loss-based retrieval in MIR has significant gains for the challenging Split-CIFAR100 setting. However, CoPE surpasses both GSS and MIR for all three benchmarks, and on top of that operates profusely more resource efficient as discussed in Appendix C. The balancing memory scheme in CoPE-CE highly improves Reservoir over imbalanced Split-MNIST and Split-CIFAR10 variants with 10.8% and 3.4% respectively, and performs on par for Split-CIFAR100 where balancing over 100 classes with limited batch size proves more difficult. Although CoPE and CoPE-CE share memory and retrieval schemes, the prototypical CoPE surpasses the cross-entropy based CoPE-CE with 4.0%, 2.9% and 6.7% respectively on the three benchmarks, indicating the merits of the PPP-loss and continually evolving prototypes. Figure 6 compares the CoPE and CoPE-CE confusion matrices at the end of learning, showing that CoPE better preserves the recall over early learned classes. CoPE-CE exhibits high plasticity as classes 8 and 9 of the last task have high recall compared to the earlier learned classes. Hence, CoPE seems to better preserve stability, effectively alleviating catastrophic forgetting.

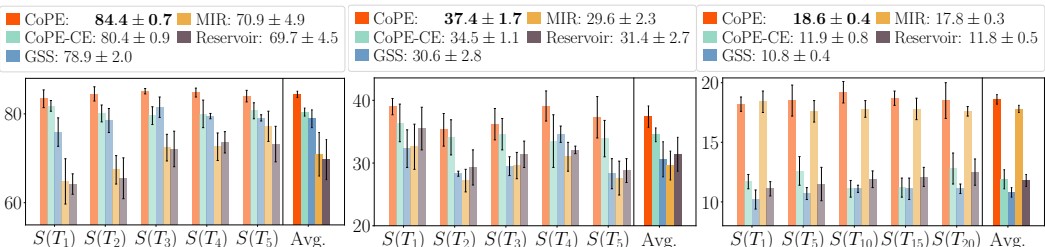

Figure 5: Accuracy (%) for imbalanced Split-MNIST (*left*), Split-CIFAR10 (*center*) and Split-CIFAR100 (*right*) sequences. The legend reports average accuracies over all the sequence variations.

## 6.3 PPP-LOSS ANALYSIS

In the challenging setting for online processing of non-iid data streams, the PPP-loss exploits information in the small processing batch $B$, introducing pseudo-prototypes $\hat{\mathbf{p}}$ on top of the prototypes. This leads to questioning to what extent the pseudo-prototypes actually contribute to the quality of the embedding, and how this relates to the batch size. We examine both inquiries in Table 2 for the

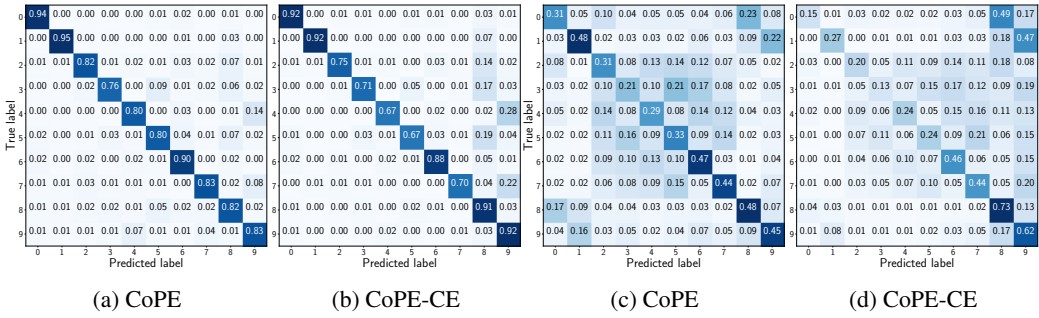

(a) CoPE      (b) CoPE-CE      (c) CoPE      (d) CoPE-CE

Figure 6: CoPE and CoPE-CE confusion matrices at the end of learning averaged over all variations $S(T_i)$ for the imbalanced Split-MNIST setup in (a) and (b), and Split-CIFAR10 in (c) and (d).

three balanced data streams by comparing inclusion and exclusion of the pseudo-prototypes $\hat{\mathbf{p}}$ in the PPP-loss, and extending the batch size $|B_n|$. First, including the pseudo-prototypes significantly improves overall performance, and especially for the harder CIFAR-based data streams. Although both setups use batch information to update the prototypes following Eq.(1), it seems crucial to use additional pseudo-prototypes in the PPP-loss to improve latent space quality. Second, results for smaller batch sizes of 10 and 20 are very similar, and deteriorate towards increasing sizes. The PPP-loss implements the expectation over the prototype and the pseudo-prototypes, assuming uniform distribution in Eq.(3). Although this assumption impedes significance of the prototype for increasingly higher batch sizes, it results in ideal robustness for small online processing batches, ideally suited for data incremental learning. Small batches maintain the additional benefit of more frequent prototype updates for the same amount of processed data.

Table 2: Accuracies (%) for ablating pseudo-prototypes $\hat{\mathbf{p}}$ in the PPP-loss and varying batch size.

| | PPP-loss | | Batch Size $|B_n|$ | | | | |
|---|---|---|---|---|---|---|---|
| | *incl.* $\hat{\mathbf{p}}$ | *excl.* $\hat{\mathbf{p}}$ | 10 (Online) | 20 | 50 | 100 | 200 |
| Split-MNIST | $93.9 \pm 0.2$ | $92.4 \pm 0.6$ | $93.9 \pm 0.2$ | $93.9 \pm 0.6$ | $93.7 \pm 0.3$ | $93.1 \pm 0.6$ | $89.3 \pm 0.5$ |
| Split-CIFAR10 | $48.9 \pm 1.3$ | $41.3 \pm 2.0$ | $48.9 \pm 1.3$ | $48.4 \pm 1.9$ | $43.4 \pm 2.7$ | $37.4 \pm 3.0$ | $37.0 \pm 1.3$ |
| Split-CIFAR100 | $21.6 \pm 0.7$ | $16.3 \pm 0.7$ | $21.6 \pm 0.7$ | $21.7 \pm 0.7$ | $16.5 \pm 0.4$ | $13.8 \pm 0.5$ | $11.2 \pm 0.4$ |

## 7   CONCLUSION

In this work, we introduced a new perspective on current paradigms in continual learning with a novel two-agent learner-evaluator framework. To overcome the standard paradigm of static training and testing phases, we explicitly model continual optimization and evaluation in the *learner* and *evaluator* agents respectively. We formalized the required resources as the *horizon* $\mathcal{D}$, containing the simultaneously available data of the data stream, and the *operational memory* $\mathcal{M}$ for operation of the learning algorithm. Transitions in the horizon $\mathcal{D}_t \rightarrow \mathcal{D}_{t+1}$ enable a uniform differentiation between existing paradigms of task, class and domain incremental learning, and the horizon size encloses the range from online ($\mathcal{D} = B$) to offline ($\mathcal{D} = S$) learning.

Using the framework, we defined the task-free *data incremental learning* paradigm, requiring no additional information on the identifier $t$ of the horizon for both the learner and evaluator. In this challenging setup, we proposed Continual Prototype Evolution (CoPE) as a prototypical solution to learn online from non-stationary data streams. As a first, CoPE prevents the prototypes becoming obsolete in an ever evolving representation space, while using the prototypes to combat catastrophic forgetting. The three main components, continually evolving prototypes, a novel Pseudo-Prototypical Proxy loss (PPP-loss), and an efficient balancing replay scheme are proven remarkably effective over 11 baselines in both balanced and highly imbalanced benchmarks. We hope to encourage research in the direction of data incremental learning with online processing of data streams and applications beyond classification.

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

APPENDIX

## A ALGORITHM

Our proposed algorithm is fully formalized in this section, as well as in our code that will be made publicly available on acceptance of this paper. Algorithm 1 and Algorithm 2 describe the learner for CoPE, whereas the evaluator uses $c^* = \arg\max_{c \in \mathcal{Y}} \mathbf{f}_i^T \mathbf{p}^c$, classifying $\mathbf{x}_i$ as category $c^*$ with the most similar prototype $\mathbf{p}^{c^*}$. As for a true continually progressing system, the evaluator can urge prediction at any point in time, while the learner keeps acquiring knowledge from the data stream.

---

**Algorithm 1** The CoPE learner in the data incremental learning setup.

---

**Require:** data stream $S$, prototype momentum $\alpha$, memory capacity $M$, learning rate $\eta$
**Initialize** operational memory $\mathcal{M} = \emptyset$, observed classes $\mathcal{Y} = \emptyset$, sample count per class $N = \emptyset$,
    model parameters $\theta$

1: **for** $B_n = \{(\mathbf{x}_1, \mathbf{y}_1), ..., (\mathbf{x}_{|B_n|}, \mathbf{y}_{|B_n|})\} \sim S$ **do**      ▷ Data stream batch w/o task information
2:      $B_\mathcal{M} \leftarrow$ RANDOMSAMPLE$(\mathcal{M}_r, |B_n|)$      ▷ Randomly sample $|B_n|$ exemplars from $\mathcal{M}_r$
3:      $\mathcal{B} = \emptyset$
4:      **for** $(\mathbf{x}_i, \mathbf{y}_i) \in B_n \cup B_\mathcal{M}$ **do**
5:          **if** $y_i \notin \mathcal{Y}$ **then**
6:              INITCLASS$(\mathcal{M}, N, \mathcal{Y}, y_i)$      ▷ Initialize memory and prototype
7:          **end if**
8:          $\mathcal{B} \leftarrow \mathcal{B} \cup f_\theta(\mathbf{x}_i)$      ▷ Collect features
9:      **end for**
10:      $\mathcal{L} \leftarrow 0$      ▷ Initialize loss
11:      **for** $\mathbf{f}_i^c \in \mathcal{B}$ **do**
12:          $\mathcal{L} \leftarrow \mathcal{L} - \frac{1}{|\mathcal{B}|}\left[\log P(c|\mathbf{x}_i^c) + \sum_{\mathbf{x}_j^k} \log(1 - P(c|\mathbf{x}_j^k))\right]$      ▷ Sum al instances PPP-loss
13:      **end for**
14:      $\theta \leftarrow \theta + \eta \nabla\mathcal{L}$      ▷ Optimize objective with SGD
15:      PROTOTYPEUPDATE$(\mathcal{M}_p, \mathcal{B}, N, \alpha)$      ▷ Update prototypes in $\mathcal{M}_p$
16:      MEMORYUPDATE$(\mathcal{M}_r, B_n, N)$      ▷ Update memory $\mathcal{M}_r$ with new input samples
17: **end for**

---

**Algorithm 2** Memory Management of the replay memory and prototypes. UNIFORM$^{\mathbb{R}^d}(s_1, s_2)$ samples elements in a $d$-dimensional vector with uniform probability in range $[s_1, s_2] \in \mathbb{R}$.

---

**Require:** memory capacity $M$

1: **function** INITCLASS$(\mathcal{M}, N, \mathcal{Y}, y)$
2:      $N \leftarrow N \cup \{N^y = 0\}$      ▷ Sample counts
3:      $\mathcal{Y} \leftarrow \mathcal{Y} \cup \{y\}$      ▷ Observed classes
4:      $m = M/|\mathcal{Y}|$      ▷ Capacity per class
5:      **for** $\mathcal{M}_r^c = (\mathbf{x}_1, ..., \mathbf{x}_{|\mathcal{M}_r^c|}) \in \mathcal{M}_r$ **do**
6:          $\mathcal{M}_r^c \leftarrow (\mathbf{x}_1, ..., \mathbf{x}_m)$      ▷ Keep first $m$
7:      **end for**
8:      $\mathcal{M} \leftarrow \mathcal{M} \cup \{\mathcal{M}^y = \emptyset\}$
9:      $\mathbf{p}^y \leftarrow$ UNIFORM$^d(0, 1)$
10:      $\mathcal{M}_p^y \leftarrow \{\mathbf{p}^y/||\mathbf{p}^y||_2\}$      ▷ Init prototype
11: **end function**

1: **function** PROTOTYPEUPDATE$(\mathcal{M}_p, \mathcal{B}, N, \alpha)$
2:      **for** $\mathbf{p}^c \in \mathcal{M}_p$ **do**
3:          $N^c \leftarrow N^c + |\mathcal{B}^c|$
4:          $\bar{\mathbf{p}}^c = \frac{1}{|\mathcal{B}^c|}\sum_{\mathbf{f}^c \in \mathcal{B}^c} \mathbf{f}^c$
5:          $\mathbf{p}^c \leftarrow \alpha\mathbf{p}^c + (1 - \alpha)\bar{\mathbf{p}}^c$
6:          $\mathbf{p}^c \leftarrow \mathbf{p}^c/||\mathbf{p}^c||_2$      ▷ Normalize
7:      **end for**
8: **end function**
9: **function** MEMORYUPDATE$(\mathcal{M}_r, B_n, N)$
10:      **for** $\mathbf{x}_i^c \in B_n$ **do**      ▷ Class Reservoir
11:          $j =$ UNIFORM$^{\mathbb{N}^1}(1, N^c)$
12:          **if** $j \leq |\mathcal{M}_r^c|$ **then**
13:              $\mathcal{M}_r^c[j] \leftarrow \mathbf{x}_i^c$ ▷ Replace exemplar
14:          **end if**
15:      **end for**
16: **end function**

---

## B  SETUP

A gridsearch in the online continual learning setup was adopted, selecting the setup with highest performance, similar to (Lopez-Paz & Ranzato, 2017). All methods are prone to learning rate gridsearch $[0.05, 0.01, 0.005, 0.001]$. iCaRL knowledge distillation strength is set to 1, and GEM bias is set to 0.5, following (Lopez-Paz & Ranzato, 2017; Rebuffi et al., 2017; Aljundi et al., 2019b). GSS and MIR follow their original setup from their codebase in (Aljundi et al., 2019b) and (Aljundi et al., 2019a), with our additional learning rate gridsearch. CURL (Rao et al., 2019) and DN-CPM (Lee et al., 2020) results, and the best imbalanced Split-MNIST results out of the greedy/IQP versions for GSS (Aljundi et al., 2019b) are reported from their original works. CoPE searched for a suitable temperature $\tau = [0.1, 0.2, ..., 1, 2]$ which was set to 0.1 for all balanced and imbalanced Split-MNIST and Split-CIFAR10 experiments, similar to (Ye et al., 2019). Based on the ablation study in Appendix D, we set the prototypical momentum fixed to 0.99. For the challenging Split-CIFAR100 setting methods are allowed multiple iterations per batch as in (Lopez-Paz & Ranzato, 2017), from which the best results are selected (baselines, reservoir, CN-DPM perform 1 iteration, others 5). The CIFAR100 temperature required higher concentration with $\tau = 0.05$ and prototypical momentum 0.9. For the balanced setups, the latent dimensionality $d$ is fixed to 100 for Split-MNIST as in (Rao et al., 2019), and selected 256 in a gridsearch $[128, 256]$ and $[128, 256, 512]$ for Split-CIFAR10 and Split-CIFAR100 respectively. The imbalanced benchmarks follow the low capacity setup in Appendix E.1, with $d \in [16, 32, 64]$ set to 64 for Split-MNIST and $d \in [128, 256]$ set to 128 for Split-CIFAR10 and 256 for Split-CIFAR100. Results are obtained without L2 normalization of the prototypes as we found it to have insignificant effect. The CIFAR10 labels in the confusion matrices from 0 to 10 stand for the indices in the following list: [*airplane, automobile, bird, cat, deer, dog, frog, horse, ship, truck*]. We will make our code publicly available upon acceptance of this paper to ensure reproducibility.

## C  RESOURCE ANALYSIS TASK-FREE REPLAY METHODS

In this section we compare usage of computational and memory resources for the replay methods fitted for the online data incremental learning paradigm.

**Reservoir** is a powerful baseline for balanced data streams (Chaudhry et al., 2019), with only minimal computational cost by keeping count $n$ of how many samples have been observed. This count is then used relative to the buffer size $M$ to define the probability $M/n$ to store the new sample. As shown in the imbalanced data stream experiments, Reservoir is not fit for more real-world scenarios with typically varying frequency of occurrence per class. Improving this simple experience replay has led to research focusing on more complex strategies, discussed in the following.

**MIR** (Aljundi et al., 2019a) replaces the random retrieval from the buffer in Reservoir with a loss-based approach. They store a momentary update of the network optimized for the new incoming batch and calculate the change in loss for a random subset of replay memories $\tilde{B}$, which is larger than the batch size (ideally five times the batch size for their experiments (Aljundi et al., 2019a)). Besides a copy of the full model, this also requires calculating the loss twice in a sequential manner for the full subset $\tilde{B}$ and an extra temporary model update using only the new batch $B_n$, both significantly increasing processing time for the learner.

**GSS** (Aljundi et al., 2019b) resides with Reservoir to use random retrieval of the buffer, but proposes a gradient-based population strategy. They introduce two variants, in which the first solves an Integer Quadratic Problem (IQP) with polynomial complexity w.r.t. the replay memory. As this is not scalable, they also propose a stochastic GSS-greedy variant. This more efficient GSS-greedy approach requires an additional forward pass, loss calculation, and backwards pass to obtain the gradients for the full considered subset $\tilde{B}$ in the memory. Additionally, it uses similarities of the gradients for stochastic sample selection in the replay memory $\mathcal{M}_r$, straining memory requirements as batch $B_n$ requires for each sample $|\tilde{B}| + 1$ gradients to be accessed simultaneously to calculate $|\tilde{B}|$ cosine similarities in the high-dimensional gradient-space.

**CoPE (ours)** resembles Reservoir's memory population by keeping count of the samples per class-specific replay memory subset. The PPP-loss requires calculation of a similarity matrix with all the $d$-dimensional representations in the batch $B$. Using a normalized cosine similarity, this implies

efficient matrix multiplication with the low-dimensional vectors. This is in high contrast to GSS, which calculates cosine similarity in the full high-dimensional gradient space for additional samples that are not present in current batch $B$, and therefore requires additional costly forward and backward passes. Furthermore, in our prototypical approach the prototype momentum updates also rely solely on samples that are in the current batch $B$, hence requiring only minimal additional computation. Comparing to both MIR and GSS, we don't require storing model copies or additional gradients, but merely store low-dimensional prototypes for each class, saving a significant amount of required storage space. For example, a Resnet18 model requires 11.7 million parameters to enable model copies or gradients, whereas our method even for 1000-way classification with $d = 1024$ would require only 9% of the model capacity in memory for the prototypes.

# D   EXTENDED ABLATION STUDY

## D.1   ABLATION PROTOTYPE MOMENTUM

In all experiments, a high momentum is employed to update prototypes with the latent mean of the batch. Table 3 illustrates the influence of higher momentum ($\geq 0.9$). Compared to low momentum of $0.1$, Split-MNIST only gains a small margin of $0.45\%$, whereas Split-CIFAR10 and Split-CIFAR100 significantly improve with at least $3.0\%$ and $4.2\%$ respectively. Using momentum prevents the prototype to rely solely on the current batch instances, and higher momentum values attain a more gradual change of the prototypes by stabilizing its trajectory in the ever-evolving latent space.

Table 3: Ablation study changing momentum strength for prototype updates, reported in average accuracy (%) over 5 runs. Higher momentum values ($\geq 0.9$) obtain better performance, especially for the CIFAR sequences, compared to low momentum ($0.1$).

|  | **Prototype Momentum** | | | |
|---|---|---|---|---|
|  | 0.1 | 0.9 | 0.95 | 0.99 |
| Split-MNIST | $93.49 \pm 0.70$ | $94.11 \pm 0.34$ | $93.96 \pm 0.30$ | $93.94 \pm 0.20$ |
| Split-CIFAR10 | $44.48 \pm 3.19$ | $48.02 \pm 2.49$ | $47.98 \pm 3.14$ | $48.92 \pm 1.32$ |
| Split-CIFAR100 | $15.79 \pm 1.16$ | $21.62 \pm 0.69$ | $21.56 \pm 0.58$ | $20.01 \pm 1.81$ |

## D.2   ABLATION INTER AND INTRA-CLASS VARIANCE TERMS PPP-LOSS

In this section, the importance is scrutinized of the two loss components to enhance inter and intra-class variance in the PPP-loss. Table 4 compares using only positive pairs from the batch in the attractor ($\mathcal{L}_{pos}$) or only negative pairs in the repellor ($\mathcal{L}_{neg}$) to the full-fledged PPP-loss ($\mathcal{L}$). The attractor term shows competitive performance to the full PPP-loss for Split-MNIST, but deteriorates as the data streams become harder for the CIFAR setups. The repellor term is on par with the full PPP-loss for Split-MNIST and Split-CIFAR10, but collapses for Split-CIFAR100. The latter is challenging due to the high number of classes with only a batch size of 10, which impedes having pseudo-prototypes of all classes in the same batch. The PPP-loss incorporates both reduction of intra-class variance with the attractor term and increases inter-class variance with the repellor term, attaining state-of-the-art performance.

Besides isolating the attractor and repellor terms of the PPP-loss in the ablation study, we further investigate the weighing of the two terms during the lifetime of the learner in Figure 7. We average results over 5 runs for balanced Split-MNIST, finding the repellor to dominate. This trend is to be expected as the repellor term in Eq.(5) has per instance a summation over all other class instances. The attractor term has minimal influence especially for data presented for the first task. This indicates the samples in the binary latent space (having observed only two classes) majorly repelling rather than attracting samples. The embedding network is still learning the initial features, and overlap in the two latent class distributions summed over the other class samples results in a prevailing repellor term.

Table 4: Ablation using solely the attractor ($\mathcal{L}_{pos}$) or repellor ($\mathcal{L}_{neg}$) compared to using both terms in the PPP-loss ($\mathcal{L}$).

| | PPP-loss | | |
|---|---|---|---|
| | $\mathcal{L}$ | $\mathcal{L}_{pos}$ | $\mathcal{L}_{neg}$ |
| Split-MNIST | $\mathbf{93.94 \pm 0.20}$ | $93.25 \pm 0.22$ | $\mathbf{93.84 \pm 0.48}$ |
| Split-CIFAR10 | $\mathbf{48.92 \pm 1.32}$ | $30.96 \pm 3.58$ | $\mathbf{49.30 \pm 3.57}$ |
| Split-CIFAR100 | $\mathbf{21.62 \pm 0.69}$ | $15.85 \pm 0.34$ | $9.43 \pm 0.94$ |

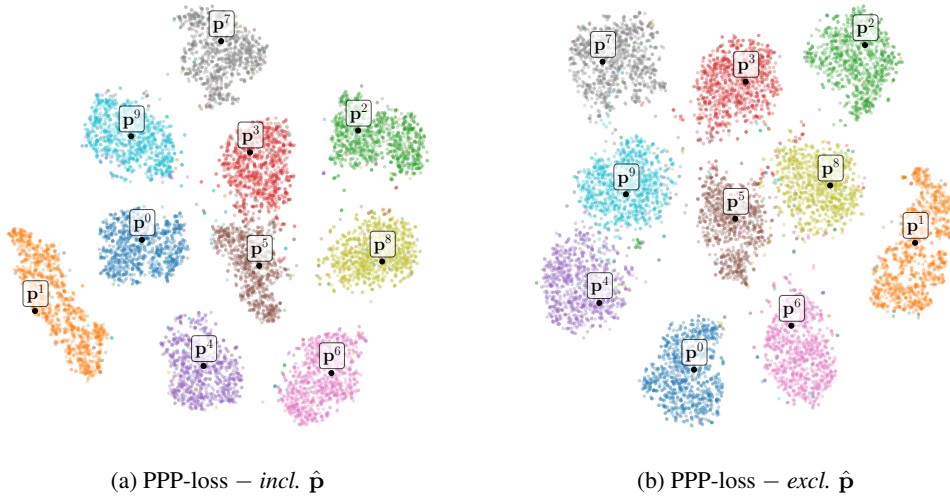

Figure 7: Weighing (%) between the positive loss term $\mathcal{L}_{pos}$ compared to the full PPP-loss $\mathcal{L}$, averaged over 5 runs of balanced Split-MNIST with standard deviation in blue.

### D.3 PSEUDO-PROTOTYPE ABLATION VISUALIZATION

In the main paper we find in an ablation study that using pseudo-prototypes $\hat{\mathbf{p}}$ as proxy for the class-mean has significant improvements for the PPP-loss. Additionally, Figure 8 shows this in a 2-dimensional t-SNE space for the first seed of the balanced Split-MNIST experiment. Including the pseudo-prototypes (*incl.* $\hat{\mathbf{p}}$) illustrates a striking degree of inter-class variance in Figure 8a, whereas more interference occurs when excluding the pseudo-prototypes in Figure 8b. This is reflected in the performance, as including prototypes results in $94.52\%$ accuracy, whereas excluding them has only $90.86\%$ for the first seed.

    (a) PPP-loss $-$ *incl.* $\hat{\mathbf{p}}$             (b) PPP-loss $-$ *excl.* $\hat{\mathbf{p}}$

Figure 8: Split-MNIST first seed t-SNE representation of the test data $S_{eval}$, including (a) and excluding (b) the pseudo-prototypes $\hat{\mathbf{p}}$ in the PPP-loss.

## E ADDITIONAL EXPERIMENTS

### E.1 BALANCED DATA STREAMS WITH LOW CAPACITY

In these experiments we scrutinize performance of CoPE with less capacity in the memory and model, and with shorter data streams. All methods are allowed multiple iterations (maximal 5) as in (Aljundi

et al., 2019b). Results are averaged over 5 seeds. Similar to the setup of GSS (Aljundi et al., 2019b), we adopt two data sequences with truncated data per task:

- **Split-MNIST-mini** is similar to the Split-MNIST data stream with 5 tasks, but each task is confined to 1k training samples. Evaluation considers the full test subset. The network is an MLP with two hidden layers of 100 units, with total memory size of 0.3k exemplars. Latent dimensionality $d$ is selected 32 from $[16, 32, 64]$.
- **Split-CIFAR10-mini** is similar to the Split-CIFAR10 data stream with 5 tasks, but each task comprises 2k training samples, with a total subset of 10k samples out of the 50k available. The full test subset is used for evaluation. The network used is the same ResNet18 as in the main paper, with total memory size of 1k exemplars. Latent dimensionality $d$ is selected 128 from $[128, 256]$.

**Analysis.** Table 5 shows the results for Split-MNIST-mini and Split-CIFAR10-mini, with GSS and DN-CPM results reported from their original works in a corresponding setup. In Split-MNIST-mini our method approaches the iid-online baseline up to $1\%$, and outperforms its closest competitors GEM and MIR with at least $1.45\%$. In Split-CIFAR10-mini CoPE saliently surpasses the iid-online baseline with $2.25\%$, hence outperforming online training over an iid datastream. Moreover, CoPE surpasses CN-DPM by $3\%$. Reservoir proves a strong baseline, with in this case the additional MIR loss-based retrieval decreasing performance. Similar to our findings in the main paper and Aljundi et al. (2019a;b), GEM encounters difficulties in a CIFAR10 based setup, for which we find the bias hyperparameter $\gamma \geq 0$ in the gradient projection to have insignificant influence. These results confirm CoPE outperforming both GSS and CN-DPM in this low capacity setting established in their original work.

Table 5: Split-MNIST-mini and Split-CIFAR10-mini results, with respectively only 1k and 2k samples per task. GSS and DN-CPM results reported from original work in these setups.

|  | Split-MNIST-mini | Split-CIFAR10-mini |
|---|---|---|
| iid-offline | $94.58 \pm 0.17$ | $67.41 \pm 1.37$ |
| iid-online | $87.57 \pm 3.54$ | $42.50 \pm 2.15$ |
| finetune | $21.74 \pm 3.38$ | $16.65 \pm 0.24$ |
| GEM | $85.09 \pm 0.52$ | $22.31 \pm 1.37$ |
| iCaRL | $83.23 \pm 0.92$ | $26.54 \pm 2.73$ |
| DN-CPM (Lee et al., 2020) | $-$ | $41.78$ |
| reservoir | $82.73 \pm 2.39$ | $38.21 \pm 3.39$ |
| MIR | $84.40 \pm 0.91$ | $37.20 \pm 2.74$ |
| GSS (Aljundi et al., 2019b) | $82.60 \pm 2.90$ | $33.56 \pm 1.70$ |
| CoPE | $\mathbf{86.54 \pm 1.41}$ | $\mathbf{44.75 \pm 2.68}$ |

### E.2 BUFFER SIZE ANALYSIS: SPLIT-CIFAR100

The results for Split-CIFAR10 and Split-MNIST are reported in the main paper, whereas Split-CIFAR100 results are added here in Figure 9 due to lack of space. We observe the same trend, where CoPE prevails over other replay methods by high margin from low to high-capacity regimes. The performance of the learner in CoPE scales with the size of $\mathcal{M}_r$.

### E.3 UNABALANCED BENCHMARK RESULTS

The graphs in the main paper visualize the numbers in Table 6, which we fully report here as a reference for future work. Each $S(T_i)$ data stream performance is averaged over five different initial seeds. The 'Avg.' results average over all mean performances of the dataset variants $S(T_i)$.

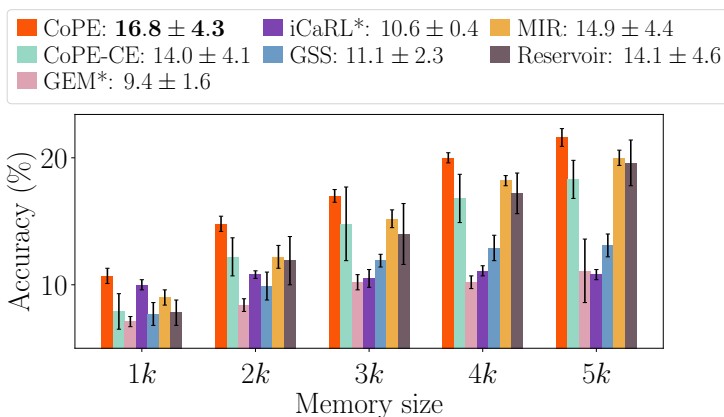

Figure 9: Accuracies over buffer sizes $|\mathcal{M}|$ for balanced Split-CIFAR100 sequence.

Table 6: Numeric results for imbalanced Split-MNIST, Split-CIFAR10 and Split-CIFAR100 sequences.

| Dataset | Imbalanced Sequence | CoPE | CoPE-CE | GSS | MIR | Reservoir |
|---|---|---|---|---|---|---|
| **Split-MNIST** | $S(T_1)$ | $83.4 \pm 2.0$ | $81.8 \pm 1.2$ | $75.9 \pm 3.2$ | $64.8 \pm 5.1$ | $64.2 \pm 2.3$ |
| | $S(T_2)$ | $84.5 \pm 1.6$ | $80.1 \pm 1.9$ | $78.5 \pm 2.7$ | $67.4 \pm 3.2$ | $65.5 \pm 4.6$ |
| | $S(T_3)$ | $85.1 \pm 0.6$ | $79.6 \pm 2.0$ | $81.5 \pm 2.3$ | $72.4 \pm 3.0$ | $72.1 \pm 4.0$ |
| | $S(T_4)$ | $84.8 \pm 1.0$ | $80.0 \pm 3.1$ | $79.5 \pm 0.6$ | $72.6 \pm 3.1$ | $73.6 \pm 2.4$ |
| | $S(T_5)$ | $84.0 \pm 1.3$ | $80.7 \pm 1.8$ | $79.1 \pm 0.7$ | $77.2 \pm 3.4$ | $73.2 \pm 4.0$ |
| | Avg. | $\mathbf{84.4 \pm 0.7}$ | $80.4 \pm 0.9$ | $78.9 \pm 2.0$ | $70.9 \pm 4.9$ | $69.7 \pm 4.5$ |
| **Split-CIFAR10** | $S(T_1)$ | $39.0 \pm 1.3$ | $36.4 \pm 3.0$ | $32.3 \pm 3.0$ | $32.6 \pm 3.6$ | $35.5 \pm 3.4$ |
| | $S(T_2)$ | $35.3 \pm 2.6$ | $34.1 \pm 2.8$ | $28.3 \pm 0.4$ | $27.2 \pm 1.8$ | $29.3 \pm 2.8$ |
| | $S(T_3)$ | $36.2 \pm 2.5$ | $34.6 \pm 2.5$ | $29.5 \pm 1.5$ | $29.6 \pm 2.1$ | $31.4 \pm 2.1$ |
| | $S(T_4)$ | $39.1 \pm 2.4$ | $33.5 \pm 4.2$ | $34.6 \pm 1.3$ | $31.0 \pm 2.3$ | $32.1 \pm 0.6$ |
| | $S(T_5)$ | $37.3 \pm 3.3$ | $33.9 \pm 2.9$ | $28.3 \pm 2.4$ | $27.6 \pm 2.7$ | $28.8 \pm 1.9$ |
| | Avg. | $\mathbf{37.4 \pm 1.7}$ | $34.5 \pm 1.1$ | $30.6 \pm 2.8$ | $29.6 \pm 2.3$ | $31.4 \pm 2.7$ |
| **Split-CIFAR100** | $S(T_1)$ | $18.2 \pm 0.6$ | $11.7 \pm 0.6$ | $10.2 \pm 0.8$ | $18.4 \pm 0.9$ | $11.1 \pm 0.6$ |
| | $S(T_5)$ | $18.5 \pm 1.3$ | $12.6 \pm 1.2$ | $10.7 \pm 0.5$ | $17.6 \pm 0.9$ | $11.5 \pm 1.4$ |
| | $S(T_{10})$ | $19.2 \pm 0.9$ | $11.1 \pm 0.7$ | $11.1 \pm 0.3$ | $17.8 \pm 0.7$ | $11.9 \pm 0.7$ |
| | $S(T_{15})$ | $18.7 \pm 0.6$ | $11.2 \pm 0.8$ | $11.1 \pm 0.9$ | $17.8 \pm 0.9$ | $12.1 \pm 0.8$ |
| | $S(T_{20})$ | $18.5 \pm 1.5$ | $12.8 \pm 1.3$ | $11.1 \pm 0.4$ | $17.6 \pm 0.4$ | $12.5 \pm 1.1$ |
| | Avg. | $\mathbf{18.6 \pm 0.4}$ | $11.9 \pm 0.8$ | $10.8 \pm 0.4$ | $17.8 \pm 0.3$ | $11.8 \pm 0.5$ |

