# OpenReview forum: "Continual Prototype Evolution: Learning Online from Non-Stationary Data Streams"
_ICLR.cc/2021/Conference — Reject_

### Official Review · AnonReviewer4 · 2020-10-28
**An interesting framework and model for learning on non-stationary data streams**

**Rating:** 8
**Confidence:** 4

**Review:**

Summary:

This article introduces a learner-evaluator framework that incorporates the different variations of problems related to incremental learning. It also proposes a method called Continual Prototype Evolution for dealing with the most general version of the problem, incremental learning on data streams, in which the learning task is not specified. The paper presents an extensive amount of experiments indicating that in this scenario, the proposed method improves significantly on existing approaches in terms of accuracy and memory efficiency. The article is well organized, easy to read and understand.

Positive aspects:

- It presents an adequate coverage of the literature on continual learning in section 3 and an interesting framework organizing the area in section 2.
- The method does not need information about the task being learned, being more applicable to real-world scenarios.
- An extensive experimental evaluation of the proposed method and comparisons with related methods is provided.
- Ablation studies indicate the most important aspects of the proposed model.

Points to discuss/improve:

- Momentum parameter: It took me a while to understand why the authors use a “high momentum”. While in gradient descent the momentum creates a tendency to keep the parameter changing in the previous directions of motion, here the momentum is supposed to make it change more slowly. There is another way of formulating Eq. 1, in which alpha works as a learning rate. Defining alpha = (1-alpha), a slow learning rate instead of high momentum, we can have Eq. 1 as: p^c = p^c + alpha(p^c - \bar{p}^c). This is easier to understand in my opinion, but authors can choose to disregard this if they prefer the current form.

- The method assumes that one prototype for each class is enough. However, in certain problems, a class might need to be represented by different prototypes, indicating the different ways of being from the same class. How the model would handle these situations?

- The conclusion does not discuss adequately the main findings of the article, probably due to lack of space.

- Page 15 is completely black in most PDF viewers I tried. Only Chrome was able to display it correctly.

Conclusion:

I believe this article should be accepted as a see it presents interesting findings in the area of incremental learning on non-stationary data streams.

---

> ### Author Response · Authors · 2020-11-12
> **Re: An interesting framework and model for learning on non-stationary data streams**
>
> Dear reviewer, we express our gratitude for your valuable evaluation of our work. We answer the points of discussion (A) below.
>
> **A1:** We prefer the usage of the term ‘momentum’ as the learning rate could cause ambiguity for the reader w.r.t. the learning rate in the optimization process. Nonetheless, we will explicitly elaborate in the discussion of our method on the semantics of ‘high’ momentum to avoid confusion.
>
> **A2:**  It is an interesting discussion about whether a single prototype per class suffices or not. We believe it does, as the prototypes are learned (i.e., they do not correspond to a particular input image) and the feature space is learned simultaneously.
> Suppose you have a class with two distinctive subcategories (say male and female individuals of a given bird species).  At first, one might think you need a prototype for each, as the two subcategories may be mapped to different areas in feature space. However, one can always add one (or a few) extra projection layers that map these distinct areas onto the same point in a higher-level representation. So if the network is deep enough, a single prototype is likely to be sufficient.
> This assumption is also supported in meta-learning (e.g. prototypical networks) and nearest mean classification (e.g. in iCaRL). However, if we would consider using multiple prototypes, our PPP-loss is designed to rely both on prototypes and pseudo-prototypes, so we could readily extend the repellor and attractor sets (see Section 4.3)  by adding the additional prototypes.
>
> **A3:**  Further, as suggested, we will extend the conclusion using the additional rebuttal page to elaborate on our main findings and we will include an overviewing figure of the framework.
>
> Please let us know if you have any more questions we can help with.

---

### Official Review · AnonReviewer3 · 2020-10-29
**Recommendation to accept**

**Rating:** 7
**Confidence:** 5

**Review:**

This paper covers an interesting topic of continual learning of the stream of data. One limitation of the existing classification algorithms is their close-set assumption. In close-set methods, a predefined set of classes are considered and a model is trained on the available data from these classes, based on the assumption that test data will be driven from a similar distribution as the training data. However, most of the real-world problems are open-set problems. Open-set models should be able to learn continuously in an online manner with minimum or zero supervision. In other words, they should be able to learn new classes or update the existing classes based on the received new data on-the-fly, without forgetting the previously learned knowledge.
Pros:
In this paper, the authors provide an incremental learning approach that prevents catastrophic forgetting.
Their approach can work on both balanced and unbalanced data.
cons:
The authors need to improve the presentation of the manuscript by providing more explanation. It could be confusing for readers who are not familiar with the topic.
It would be very helpful to publically share the code.
I highly recommend adding the confusion matrix or F1 score in addition to the accuracies.

---

> ### Author Response · Authors · 2020-11-12
> **Re: Recommendation to accept**
>
> Dear reviewer, thank you for your fruitful evaluation of our work.
> The extra page will be used to provide additional explanation as suggested, with an overview figure of the framework, and we will extend the conclusion as suggested by AnonymousReviewer4. The code will be publicly released upon paper acceptance (to comply with the double-blind policy) as we also believe it to be of key importance to support the reproducibility of our results. In addition, we hope that the comprehensive overview of the setup in Appendix B and the algorithm details in Appendix A make it easier to reproduce our work. Following your suggestion, we will additionally provide the confusion matrices in the updated paper version to get further insight into the results.
> Please reach out if you have any further questions.

---

### Official Review · AnonReviewer1 · 2020-11-01
**A new heuristic for online learning from non-IID data**

**Rating:** 3
**Confidence:** 4

**Review:**

This paper proposes a new procedure for continual supervised learning from a non-IID data stream that assumes ability to maintain some of the stream examples in a buffer and use the buffer to improve updates of a prediction model. The proposed procedure is called Continual Prototype Evolution (CoPE), which controls evolution of prototypes to prevent catastrophic forgetting.
Strenghts:
- the paper presents very detailed experimental results
Weaknesses:
- the paper is not easy to read
- the underlying assumptions about the data stream are not clearly defined. Thus, it remains unclear what problem the proposed algorithm is trying to solve.
- the proposed algorithm is a collection of heuristics that are not clearly justified. There is no attempt to provide a theoretical insight about the behavior of the algorithm.
- the experiments were performed for one particular synthetic setup on one benchmark data set and the proposed algorithm is compared to a very limited class of baselines. Thus, even the empirical evaluation is not very insightful.
- there is no discussion about the computational cost
Overall rating:
This paper has too many weaknesses and is not ready for publication

---

> ### Author Response · Authors · 2020-11-12
> **Re: A new heuristic for online learning from non-IID data**
>
> Dear Reviewer, thank you for your valuable feedback. We address the raised concerns in the following answers (A).
>
> **A1:**  We will use the extra page to improve readability, mostly along the lines of suggestions made by AnonymousReviewer3.
>
> **A2:**  We are surprised by your comment that the assumptions about the data stream are not clearly defined, as this is exactly why we introduced the learner-evaluator framework in Section 2. In Continual Learning (CL) literature, these assumptions often remain implicit which makes the comparison of methods cumbersome. With this problem in mind, we propose in Section 2 a new framework that subdivides the setups within CL based on their assumptions on the data stream by disentangling information available to the learner and evaluator.
> Alternatively, maybe the reviewer refers to more general basic assumptions, such as smoothness or P(Y|X) being task-independent or stationary? These are rather standard in the field of CL, and therefore we did not write them down explicitly. To be clear, we specify our setting below:
> i) We work in the data incremental setting under our general learner-evaluator framework.
> ii) We assume the class distributions P(Y|X) to be stationary and identical for learner and evaluator, while P(X) is non-stationary for the learner and stationary for the evaluator.\
> \
> Note that the above assumptions apply to our specific setup proposed and evaluated in sections 4 and 6, while the learner-evaluator framework of Section 2 is more generic, including also the field of concept drift with non-stationary data streams for evaluation.
>
> **A3:**  We recognize theoretical justification as a major problem in deep learning, and continual learning specifically. Nonetheless, CoPE is based on well established and empirically validated methods within continual learning (experience replay), representation learning (nearest mean classifiers), and unsupervised representation learning (instance-based contrastive losses). We provide extensive ablation studies to justify each of the components constituting CoPE, such as
>      -  the benefit of using a balanced replay buffer (CoPE-CE),
>      -  additionally using a prototypical approach (CoPE vs CoPE-CE in all experiments),
>      -  the effects of including the pseudo-prototypes in the PPP-loss (Table 2),
>      -  the advantages in low batch size regimes (Table 2),
>      -  the effects of the replay-buffer size (Figure 3).
>
> Further in Appendix D, we provide further ablation studies,
>      -  the effects of the low and high momentum regimes,
>      -  isolating the repellor and attractor loss terms in the PPP-loss,
>      -  the ratio of the repellor and attractor loss terms during training.
>
> **A4:**  *“the experiments were performed for one particular synthetic setup on one benchmark data set and the proposed algorithm is compared to a very limited class of baseline”*\
> We are not sure if we understand the concern of the reviewer. The experiments were conducted for 8 benchmarks based on Split-MNIST, Split-CIFAR10, and Split-CIFAR100, and  CoPE is compared to 11 baselines. The balanced data streams are standard benchmarks in the field, separating tasks in the data stream, which enables us to compare w.r.t. state-of-the-art requiring task-boundaries (e.g. iCaRL and GEM). On top of that, we provide 3 highly imbalanced benchmarks, resembling a more real-world setup, where one task has significantly more (e.g. factor 10) data compared to other tasks. This setup additionally shows robustness of our method in class-imbalanced regimes, where we attempt to push continual learning towards more realistic (and highly imbalanced) data streams.
>
> **A5:**  *“there is no discussion about the computational cost”*\
> This discussion has been established in Appendix C  due to lack of space, but provides a detailed discussion of the computational cost compared to the current state-of-the-art replay methods.
>
> We hope this could provide some clarifications. Please reach out for any further concerns.

---

### Author Response · Authors · 2020-11-19
**CoPE: A Rebuttal Revision**

Dear area chair,
Dear reviewer,

Thank you for your valuable reviews which we have incorporated into the revised submission, with an extra page for the rebuttal version. We included additional elaboration on our framework with an overviewing figure (Figure 1, page 2), and the additional page allowed us to significantly elaborate on our main findings in the conclusion as suggested by the reviewers.
We made our notion of ‘high momentum’ explicit in Section 4.1 by indication of high alpha (AnonymousReviewer4).
Furthermore, the discussion in Section 6.2 on the imbalanced benchmarks is extended by means of confusion matrices (AnonymousReviewer3) of CoPE compared to the CoPE-CE baseline for both Split-MNIST and Split-CIFAR10, showing the efficacy of CoPE in alleviating catastrophic forgetting.

Any further elaboration is given in individually addressed comments below, where references to figures and tables refer to the original submission.

We would gladly help you out with any further questions.

---

### Decision · Program_Chairs · 2021-01-07
**Final Decision**

**Decision:**

Reject

**Comment:**

This paper presents an interesting idea for task-free incremental learning on the data stream. The reviewers have extensive discussions after reading all the reviews and the author's rebuttal. There are concerns raised about the presentation of the method and the justification for some parts of the model design choices. The reviewers believe that after addressing these weaknesses the work can be made stronger and may be accepted in a competitive venue.